# Morphological, Molecular and Pathological Characterization of *Phytophthora pseudocryptogea* Associated with *Rosmarinus officinalis* Dieback in Tuscany, Central Italy

**DOI:** 10.3390/microorganisms13030567

**Published:** 2025-03-03

**Authors:** Alessandra Benigno, Chiara Aglietti, Santa Olga Cacciola, Salvatore Moricca

**Affiliations:** 1Department of Agriculture, Food, Environment and Forestry (DAGRI), Plant Pathology and Entomology Section, University of Florence, Piazzale delle Cascine 28, 50144 Florence, Italy; chiara.aglietti@unifi.it; 2Department of Agriculture, Food and Environment, University of Catania, 95123 Catania, Italy; olgacacciola@unict.it

**Keywords:** decline, nursery stocks, infested soil, isolation techniques, pathogenicity testing

## Abstract

A severe dieback of rosemary (*Rosmarinus officinalis* L.) plants was observed in a medicinal/culinary herb plantation in Casole d’Elsa, Siena, central Italy. Symptoms included stunted growth, crown desiccation, root rot, collar rot and internal tissue necrosis, strongly indicative of *Phytophthora* root and crown rot syndrome. Morphological and molecular identification (ITS and Cox1 sequencing) of strains isolated from symptomatic stems, roots and soil revealed the occurrence of two *Phytophthora* species: *Phytophthora pseudocryptogea*, which constituted 94% of isolates obtained from the stem, root apparatus and rhizosphere; and *Phytophthora megasperma*, which was not recovered from plant organs or tissue, being exclusively isolated from rhizosphere soil samples at a low isolation rate (6%). The pathogenicity of the obtained strains was assessed by inoculating eighteen-month-old *R. officinalis* plants in a soil infestation trial. Plants inoculated with *P. pseudocryptogea* strains died 10 days after artificial inoculation. *P. pseudocryptogea* was subsequently re-isolated from the roots of inoculated, symptomatic plants, thus fulfilling Koch’s postulates. Plants inoculated with *P. megasperma* strains were in good vegetative condition and did not show any visible symptoms, suggesting *P. megasperma* to be nonpathogenic. Artificial inoculation tests thus confirmed *P. pseudocryptogea* to be the aetiological agent responsible for the death of *R. officinalis* plants in the plantation under study. This is the first report of root, collar and crown rot caused by *P. pseudocryptogea* on *R. officinalis* in Italy. There is evidence that poorly drained soils and climate constraints facilitate the spread of this oomycete. These findings highlight the critical role of nursery trade in the introduction of *Phytophthora* species in agroecosystems and emphasize the need for more stringent control measures.

## 1. Introduction

Rosemary (*Rosmarinus officinalis* L.) is a shrubby species belonging to the Lamiaceae. This family includes xeromorphic and evergreen forms that can be found throughout the Mediterranean area as wild or cultivated plants [1]. *R. officinalis* is a globally significant crop, valued for ornamental, industrial and culinary purposes. It is also rich in bioactive compounds with antibacterial, antidiabetic, anti-inflammatory, antitumor and antioxidant properties, which make it particularly useful in the medical field. Additionally, some of its natural compounds exhibit allelopathic activity, spurring research into their potential applications [2,3,4,5,6,7,8,9,10,11]. Despite being native to the Mediterranean and Asia, *R. officinalis* is now cultivated worldwide, commonly grown also in cooler areas, due to its adaptability to diverse climatic conditions, displaying tolerance to frost, though struggling to survive in oxygen-deprived environments [4,12,13]. In Italy, *R. officinalis* represents the only native plant of the genus, occurring as a polymorphic pioneer colonizer of a variety of different habitats mainly constituted by dry, sunny and calcareous open areas that can be found from sea level up to 1500 m [14]. For all these properties, it is important to safeguard this shrub species and favour its crop exploitation.

*R. officinalis* is attacked by several fungal pathogens, the more prominent being *Rhizoctonia solani* (web blight, root rot), *Sclerotinia sclerotiorum* (stem rot), *Fusarium* spp. (vascular system disruption, rapid wilt and dieback), powdery mildews (defoliation and leaf distortion) and *Colletotrichum* (anthracnose) [15,16,17,18,19,20]. However, a particularly serious threat to *R. officinalis* is represented by oomycetes of the genus *Phytophthora*, causal agents of root, collar and crown rot [21,22,23]. The threat posed by these latter pathogens can be made even more serious by water stagnation conditions that favour their spread and resistance as latent forms. In addition, the intensification of extreme weather events brought about by climate change (e.g., prolonged rainfall, flooding) coupled with the introduction of invasive species can give rise to new outbreaks and phytosanitary emergencies [24,25,26,27]. Interspecific hybrids are reported with increasing frequency among members of the genus *Phytophthora*, brought about by large genotypic and phenotypic changes. These changes could enhance these hybrids’ ability to expand their host range, providing more opportunities for transmission and survival on the new hosts they are able to infect and damage [28]. Several *Phytophthora* species have been reported on *R. officinalis* from different countries. Specifically, *R. officinalis* was reported as a host of *P. nicotianae* in Spain and the US; of *P. tropicalis* and *P. palmivora* in Taiwan; of *P. parasitica* in Greece; of *P. cryptogea* in France and Italy; of *P. pseudocryptogea* in France; of *P. citrophthora* in the US; and of *P. cinnammomi* in New Zealand [29,30,31,32,33,34,35,36,37,38]. *Phytophthora* species are frequently found in nurseries, where they commonly infest stocks of crop and ornamental plants, impairing nursery businesses and posing a risk to agricultural systems and natural ecosystems [38,39,40]. Given *R. officinalis* is a multipurpose species largely grown in nurseries as a culinary herb, a potted florist’s crop or an ornamental, the risk that dangerous species of *Phytophthora* may spread from nurseries to *R. officinalis* cultivation fields through infected *R. officinalis* plantlets is very high. In this context, both phytosanitary inspections in nurseries and monitoring of *R. officinalis* cultivation fields for the possible occurrence of harmful *Phytophthora* pathogens become indispensable. This work was born within the framework of a phytosanitary surveillance conducted in an *R. officinalis* cultivation area and reports on the aetiology of a severe dieback observed in 2023 in an *R. officinalis* plantation at Casole d’Elsa (Tuscany, central Italy), with many plants exhibiting typical symptoms of *Phytophthora* infection. To our knowledge, this represents the first documented report of *P. pseudocryptogea* causing root, collar and crown rot on *R. officinalis* in Italy. These results highlight the need for enhanced phytosanitary measures to mitigate the spread of this pathogen on rosemary in the Mediterranean, a recognized hotspot area for climate change.

## 2. Materials and Methods

### 2.1. Field Analyses and Sampling

Investigations were conducted in an agricultural company specializing in the cultivation of medicinal plants in Casole d’Elsa (43.3384489, 11.0096936), Siena, central Italy. The *R. officinalis* plants investigated were between 4 and 7 years old. Plants exhibited symptoms such as leaf chlorosis, along with a decline and decay of the foliage. Cortical lesions were identified on the collar, indicative of *Phytophthora* infestation. Sampling was conducted in January 2023, collecting portions of necrotic roots and stems from 120 symptomatic *R. officinalis* plants. Rhizosphere soil samples, including also fine roots, were collected from soil portions immediately beneath the collar of all symptomatic plants, after removing the upper organic soil layer. Soil sampling was conducted as described by Benigno et al. [41]: four soil cores were collected for each plant, 10–20 cm away from the stem base, and were bulked together (approx. volume 1 L).

### 2.2. Isolation and Morphological Identification

A total of 288 samples were collected from 120 infected plants; these samples included shoots, crowns, stems, coarse roots and fine roots. All rosemary samples were washed, surface-sterilized in 1% NaClO for 2 min, immersed in 50% ethanol for 30 s, rinsed with sterile distilled water, dried, placed on selective PARPNH V8-agar medium, and supplemented with 10 μg/L pimaricin, 200 μg/L ampicillin, 10 μg/L rifampicin, 25 μg/L pentachloronitrobenzene, 50 μg/L nystatin, and 50 μg/L hymexazol. Plates were incubated in the dark at 24 °C for 24–48 h, and pure cultures were obtained by transferring individual hyphal tips onto V8-juice agar, MEA and PDA.

Overall, 56 soil samples were collected following the protocol of Benigno et al. [41]. To isolate *Phytophthora* species, 200 mL of soil subsamples were submerged in 800 mL of distilled water in plastic containers and baited with fresh, healthy leaves of *Quercus ilex* L., *Viburnum tinus* L. and *Rosa chinensis* Jacq. Necrotic lesions appeared on the bait leaves 2–3 days post-baiting. Necrotic lesions were aseptically excised from bait leaves and placed on V8-PARPNH medium. Within 2–3 days, coenocytic, white, soft and slow-growing mycelium emerged on the selective medium. The isolates were then transferred to V8A medium (100 mL/L V8 juice, 2 g/L CaCO_3_, 20 g/L agar) following the protocol of Jung et al. [42], and grouped into two distinct morphotypes on the basis of colony phenotypes and micro-morphological characteristics. Cardinal temperatures for their growth were assessed on V8A plates incubated at 5, 10, 15, 20, 25, 30 and 35 °C (±0.5 °C) in darkness, with seven replicates per isolate; colony diameters were recorded daily for 5 days.

The reproductive system of each morphotype was assessed after 20 days of growth on CA medium at 20 °C in the dark. Morphological structures, including sporangia, chlamydospores, hyphal swellings, oogonia and antheridia, were observed and measured for each isolate using an optical microscope (DM750; Leica Microsystems, Wetzlar, Germany) equipped with LAS EZ software version 2.0. The sexual compatibility of *Phytophthora pseudocryptogea* isolates was assessed by pairing them with known A1 and A2 mating types of *P. cryptogea* present in our collection. Mycelial plugs (5 mm in diameter) from the test isolates and the reference strains were positioned 4 cm apart on 10% clarified V8 agar medium. These dual cultures were incubated in darkness at 24 °C for 10 days or until oospore formation was observed. For each isolate, sporangial morphology (*n* = 50) and the capacity to form hyphal swellings and chlamydospores were documented. Morphometric data were obtained by measuring 50 structures per sample at 400× magnification. On the basis of the above-mentioned cultural and micro-morphometric traits, the two groups of isolates were ascribed to *P. pseudocryptogea* or *P. megasperma*. Representative morphotypes were stored on CA medium in the culture collection of filamentous fungi and oomycetes held at the DAGRI Department, School of Agriculture, University of Florence, Italy.

### 2.3. Molecular Identification

All morphotypes were transferred onto sterile cellophane in 9 cm Petri dishes containing PDA and kept in the dark at 24 ± 1 °C for 1 week. Mycelium was scraped from the cellophane surface and stored in 2 mL Eppendorf tubes at −20 °C. DNA was extracted from 7-day-old cultures grown on V8 agar at 20 °C using the GenElute Plant Genomic DNA Miniprep kit following the manufacturer’s instructions (Sigma Aldrich, St. Louis, MI, USA). The extracted DNA was then stored at −20 °C. Amplification of the ITS (internal transcribed spacer) region, including ITS1 and ITS2 spacers and the 5.8S rDNA gene, was conducted by PCR using the universal primers ITS1/ITS4 designed by White et al. [43]. PCR was performed in a final volume of 25 µL containing PCR Buffer (1×), dNTP mix (0.2 mM), MgCl_2_ (1.5 mM), forward and reverse primers (0.5 µM each), Taq DNA Polymerase (1 U) and 100 ng of DNA template. The thermocycler conditions were as follows: 94 °C for 3 min; followed by 35 cycles of 94 °C for 30 s, 55 °C for 30 s, and 72 °C for 30 s; and then 72 °C for 10 min. PCR products were purified using the NucleoSpin^®^ Gel and PCR Clean-up kit (Macherey-Nagel, Düren, Germany) following the manufacturer’s instructions. Nucleotide sequences were read and edited using FinchTV version 1.4.0 (Geospiza, Inc., Seattle, WA, USA). The investigated morphotypes were identified as *P. pseudocryptogea* or *P. megasperma* by comparing their respective consensus sequences with those deposited on NCBI using the Basic Local Alignment Search Tool (BLAST; http://www.ncbi.nlm.nih.gov/BLAST, accessed on 5 June 2023), thus confirming conventional morphology-based identification. The obtained sequences were submitted and deposited in the NCBI GenBank (Table 1). Sequences were aligned with ClustalX v. 1.83 [44], using the parameters reported by Bregant et al. [45]. Maximum likelihood (ML) analyses were performed with MEGA-11 software, including all gaps in the analyses. The best model of DNA sequence evolution was determined automatically by the software [46].

### 2.4. Pathogenicity Tests

The pathogenicity of the two *Phytophthora* species was assessed through a soil inoculation trial. One isolate per species was employed in the experiment, as laboratory examinations revealed no difference in micromorphology and physiological requirements (e.g., growth/temperature relationships) among the strains. Isolate PP150432 of *P. pseudocryptogea*, obtained from the roots of symptomatic plants, and isolate PQ510851 of *P. megasperma*, collected from rhizosphere soil, were used for the inoculation tests. Eighteen-month-old *R. officinalis* plants were transplanted into plastic pots containing a mixture of autoclaved commercial organic substrate (Compo Bio, COMPO Italia Srl, Cesano Maderno, Italy) and inoculum. The *Phytophthora* inoculum was prepared by cultivating isolates for 4–6 weeks at 20 °C on fully hydrated sterilized millet seeds supplemented with V8 broth (200 mL V8 juice, 3 g CaCO_3_, 800 mL distilled water). Thirty control plants were transplanted into plastic pots containing a mixture of non-infested soil. After transplantation, all plants were maintained in saturated soil for 4 days, after which they were transferred to a growth chamber set at 24 °C, 80% relative humidity, and a photoperiod of 16 h of light followed by 8 h of darkness. The experimental phase concluded when the inoculated plants exhibited severe symptoms, including leaf yellowing, wilting, defoliation and mortality. At the end of the trial, the plants were harvested, symptoms were analyzed and re-isolations were conducted on appropriate culture media. Colony morphology assessment, micromorphology observations and sequencing of the ITS region of ribosomal DNA were employed to identify the isolates recovered from inoculated cuttings. Root biomass growth rates were also assessed following the pathogenicity tests. For this purpose, the roots of all 90 plants (60 treated and 30 untreated) were harvested and carefully rinsed to remove any soil particles. Fresh weights of the roots were determined using a precision analytical balance. Subsequently, the roots were dried in an oven at 80 °C until complete desiccation (24 h). After drying, the roots were allowed to cool at room temperature before being weighed again to obtain dry weights. Weight measurements were taken with precision, to ensure accuracy in determining the root biomass growth rates for both treated and untreated plants. The data collected were used for comparative analysis of root biomass between the different treatment groups.

### 2.5. Statistical Analyses

Pathogenicity test data were statistically compared by analysis of variance (ANOVA). Significant differences between mean values were determined using Fisher’s least significant difference (LSD) multiple-range test (*p* = 0.05) after one-way ANOVA using the software SPSS V.28 (IBM Corporate, Endicott, NY, USA). Differences achieving a confidence level of 95% were considered significant.

## 3. Results

### 3.1. Field Surveys

Symptoms were observed on a total of 120 *R. officinalis* plants on shoots, crowns, basal stems, collar roots, coarse roots and fine roots. Initial symptoms appeared as leaf chlorosis and yellowing, crown wilting and stunting; they were randomly visible on only a few scattered individuals in September 2022. By the assessment made in January 2023, approximately 70% of the plants were severely damaged, with varying degrees of symptoms including stunting, leaf yellowing, defoliation, crown wilting with a change in the colour towards ash-grey, basal stem necrosis and eventual death of the entire plant (Figure 1). Epigeal symptoms were consistently associated with severe rot of coarse and fine roots and basal taproot necrosis in the hypogeal plant portions.

### 3.2. Isolate Identification

A total of 148 *Phytophthora* isolates were recovered from 288 rosemary tissue samples and 56 soil samples. Eleven distinct morphotypes were identified from these isolates. Three morphotypes were obtained from rhizosphere soil, six from fine roots, and three from the necrotic basal stem (Table 1). Two species of *Phytophthora*, namely *P. pseudocryptogea* and *P. megasperma*, were isolated from samples taken from root tissues, rhizosphere soil and basal stem portions of symptomatic individuals. *P. pseudocryptogea* was more consistently isolated than *P. megasperma*: from a total of 11 isolates that were characterized, 10 were found to belong to *P. pseudocryptogea* and only 1 isolate belonged to *P. megasperma* (Table 1).

The recovered isolates were morphologically ascribed to two different morphotypes. Colonies of morphotype 1, identified as *P. pseudocryptogea*, exhibited a chrysanthemum appearance after 7 days of incubation on V8A, MEA and PDA. Growth tests at different temperatures showed that the minimum growth temperature for this oomycete was 5 °C, the maximum was 35 °C, and the optimum was 25 °C. Colonies of isolates of morphotype 2, ascribed to *P. megasperma*, presented a flat and regular surface topography and did not show any distinctive pattern on V8A, MEA and PDA media. Growth temperatures were somewhat different for *P. megasperma*: while its minimum growth temperature was the same as the other oomycete (5 °C), its optimum was 20 °C (5 °C lower than *P. pseudocryptogea*), and its maximum was 25 °C (10 °C lower that *P. pseudocryptogea*).

*P. pseudocryptogea* showed non-papillate, persistent, terminal sporangia, which were ovoid, ellipsoidal or pear-shaped, measuring 42.5 ± 5.6 × 25.7 ± 2.3 μm, with a length/width ratio of 1.44. Hyphal swellings were globose to irregular; no chlamydospores were observed. In dual cultures, all isolates produced gametangia and oospores only when paired with the A2 tester strain of *P. cryptogea*, confirming their classification as the A1 mating type. Oogonia were globose with an average diameter of 32 ± 3.3 μm. The antheridia were amphigynous, and the oospores were aplerotic to nearly plerotic, with an average diameter of 31.1 ± 3.9 μm.

*P. megasperma* exhibited non-papillate, persistent sporangia that were elongated, obpyriform, limoniform, or distorted in shape (54.6 ± 8.6 × 33.7 ± 1.3 μm), some with a tapered base. Sporangia showed external, internal and nested proliferation, originating from unbranched sporangiophores. Hyphal swellings were globose, subglobose or elongated, with radiating hyphae. Chlamydospores were not observed. In single culture, spherical oogonia with an average diameter of 37 ± 5.7 μm were observed, along with spherical or ellipsoidal antheridia (16.3 ± 1.1 × 14 ± 1.5 µm), predominantly paragynous. Oospores were aplerotic (33.9 ± 5.9 µm in diameter).

DNA sequencing confirmed the identity of representative morphotypes as *P. pseudocryptogea* and *P. megasperma*. BLAST searches revealed 100% homology between the isolates and reference sequences of the two species deposited in the GenBank database (Table 1). The ITS-generated sequences were edited and aligned with representative isolates from clade 8 for *P. pseudocryptogea* and from clade 6 for *P. megasperma*. Isolates formed well-supported clades, clustering with sequences from ex-type cultures (Figure 2). Additionally, isolates belonging to clades 4, 5, 7 and 9 were incorporated into the phylogenetic tree (Figure 2).

### 3.3. Pathogenicity Tests

All 30 *R. officinalis* seedlings inoculated with *P. pseudocryptogea* exhibited severe symptoms like root rot, leaf chlorosis, defoliation, crown desiccation and eventual death within two weeks post-transplant. The great majority (80%) of the plants died at 10 days from inoculation and the remaining 20% were dying or dead after 30 days. In contrast, control plants and those inoculated with *P. megasperma* were healthy-looking, showing no aerial symptoms nor necrosis of the roots.

The average severity of symptoms was significantly higher in plants inoculated with *P. pseudocryptogea* compared to control plants, with a statistical significance of *p* = 0.05. *P. pseudocryptogea* was successfully re-isolated from the roots and stem tissues of symptomatic cuttings, thereby fulfilling Koch’s postulates (Figure 3). *P. megasperma* was sporadically re-isolated from roots. The identity of re-isolated *P. pseudocryptogea* and *P. megasperma* strains was confirmed by macro- and micro-morphological analyses and ITS sequencing. A reduction in growth rate and biomass production was observed in *R. officinalis* plants inoculated with both oomycetes; however, the most substantial reduction in growth rate and biomass was recorded in individuals inoculated with *P. pseudocryptogea*. One-way ANOVA pairwise comparisons indicated significant differences (*p* < 0.05) in root biomass among the plants inoculated with the two oomycetes, and the control group (Figure 4).

## 4. Discussion

*P. pseudocryptogea* was identified as the aetiological agent responsible for the severe decline and dieback of *Rosmarinus officinalis* plants in a plantation at Casole d’Elsa, Tuscany, central Italy. Several previous studies indicated *P. pseudocryptogea* as an aggressive root pathogen of a variety of host plants, either when acting singly or in combination with other pathogens [47]. In this study, *P. megasperma* was also isolated from rhizosphere soil samples of the affected *R. officinalis* plants; however, only *P. pseudocryptogea* was able to reproduce the symptoms observed in the field on artificially inoculated plants, with most of the infected plants being found dead at the end of the experiment. The fulfilment of Koch’s postulates, with the re-isolation of *P. pseudocryptogea* from infected plant tissues, definitely established the proof of causality. *P. megasperma* appears to play a dual ecological role, persisting in an asymptomatic equilibrium with its host under non-stress conditions but spreading and probably becoming mildly pathogenic when its host undergoes stress conditions. In fact, it cannot be excluded that some fine roots may have been attacked by *P. megasperma*, although this did not result from laboratory isolations. Field detections of *P. megasperma* were always associated with *P. pseudocryptogea* infection of *R. officinalis*, suggesting a context-dependent pathogenicity. However, controlled inoculations of *P. megasperma* in the laboratory with plants in optimal health conditions and with the oomycete inoculated singly did not show any symptom expression.

Similar *R. officinalis* dieback caused by *P. pseudocryptogea* has been reported in France, while the closely related *P. cryptogea* has been reported in France and Italy. Both *Phytophthora* species were recovered from the rhizosphere of symptomatic plants [34,37,48,49]. Anyhow, this is the first report of *P. pseudocryptogea* affecting *R. officinalis* in Italy. *P. pseudocryptogea* seems to be quite a polyphagous species, with other reported hosts being members of the Aquifoliaceae, Asparagaceae, Asphodeliaceae, Cupressaceae, Fagaceae, Lauraceae, Oleaceae, Platanaceae, Proteaceae, Salicaceae and Solanaceae families [47,50,51]. Until recently, the geographical distribution of *P. pseudocryptogea* was thought to be confined to Asia, Australia and South America. However, the repeated reports of this oomycete from Italy, Spain and Turkey in recent years prove the establishment and spread of this species in the Mediterranean region [52]. There is therefore evidence that the host range of *P. pseudocryptogea* is expanding steadily, probably as a consequence of climate change. On the other hand, in this study *P. pseudocryptogea* showed a tendency towards thermotolerance.

Multigene phylogenetic analyses, performed to clarify species taxonomy, led to distinguishing *P. pseudocryptogea* from the related *P. cryptogea* and, starting from 2015, *P. pseudocryptogea* was formally separated from *P. cryptogea* sensu lato and recognized as a new species in clade 8a. However, many *Phytophthora* species, including the *P. pseudocryptogea* recently placed in clade 8a, remain poorly characterized. *Phytophthora* taxonomy is also complicated by the occurrence of hybrids. The genus *Phytophthora* contains in fact several interspecific hybrids which pose further taxonomic conundrums. Indeed, hybridization in addition to enhancing offspring’s ability to spread and infect a broader range of hosts, calls into question the need for further research as it generates intermediate forms between the parental species, that are a cause of taxonomic confusion [28,53]. *P. pseudocryptogea* is recognized as a destructive species. It has been associated with necroses in *Coleus* spp. on which it also caused a significant inhibition of root growth [54]. Additionally, it has been linked to a dieback syndrome in *Cycas revoluta* under specific conditions [47]. The pathogenicity of *P. pseudocryptogea* was also reaffirmed in this study by root weight measurement data, with the root system of inoculated plants being three times smaller than that of plants inoculated with *P. megasperma*.

The main factors contributing to the introduction, establishment/re-establishment and spread of *Phytophthora* spp., especially in containerized nursery material, are the lack of effective controls on the health of plant material in the nursery industry and the inadequate nursery practices. Currently, nurseries source plant material primarily from a limited number of large-scale producers with both national and international reach, as well as from brokers and other wholesalers or retailers. Due to extensive trade at both vertical and horizontal levels, individual plants frequently pass through multiple nurseries, domestically and internationally, with minimal regulation before reaching the field [26,38]. This system can facilitate the rapid spread of soil-borne pathogens like *Phytophthora* species, which may be distributed throughout the trade network either through fungicide-treated but asymptomatic host plants or on non-host plants serving as passive carriers [38]. The use of healthy nursery stock is thus essential to prevent *Phytophthora* diseases in commercial plantations. Among the practices that are conducive to *Phytophthora* spread in the nursery are establishing excessively dense plantings; reusing compost or plastic containers without proper sterilization; employing unfiltered surfaces; using re-circulated irrigation water without adequate filtration or sterilization; storing nursery stock in containers on poorly drained surfaces or directly on the ground; and allowing diseased plants to accumulate in the proximity of the production area [38,55,56,57]. *R. officinalis* is a species that requires water, and in the Mediterranean climate, irrigation is essential for successful planting as well as later, during hot, dry summers for encouraging vigorous growth. However, in clayey and poorly drained or water-logged soils, *Phytophthora* root rot root asphyxiation may develop. The selection of well-drained soil for new plantings and careful irrigation management are necessary for disease prevention [50]. Less drainage and clay soil have contributed to the pervasive infestation and spread of the two oomycetes in the investigated plantation.

## 5. Conclusions

It is highly likely that the *Rosmarinus officinalis* plants found infected in the plantation at Casole d’Elsa arrived contaminated from the nursery. An anamnesis aimed at verifying the possible causes of the introduction of the two species of *Phytophthora* into the *R. officinalis* plantation strongly supports this hypothesis. Indeed, the plantation owner reported having observed similar symptoms the previous year in a lavender (*Lavandula angustifolia* Mill.) plantation established on the same estate. Upon specific questioning, he confirmed that the plant material utilized in both plantations had been purchased from the same nursery.

Without effective control measures, root and collar rot of *R. officinalis* becomes visibly apparent as repentine dieback of the foliage which, in the most severe cases, leads to the death of affected branches and/or the entire plant. Since *P. pseudocryptogea* thrives under water stagnation and poor drainage conditions, choosing well-drained soils for new plantings and employing careful irrigation management are fundamental to prevent infections by this oomycete. The increase in *Phytophthora* infestations in the Mediterranean is also due to climatic vagaries: the alternation of periods of drought and excessive rainfall, increasingly frequent in the Mediterranean area due to climate change, together with particular soil conditions, like permanently or periodically wet soils, create ideal environments for the proliferation of *Phytophthora* species.

## Figures and Tables

**Figure 1 microorganisms-13-00567-f001:**
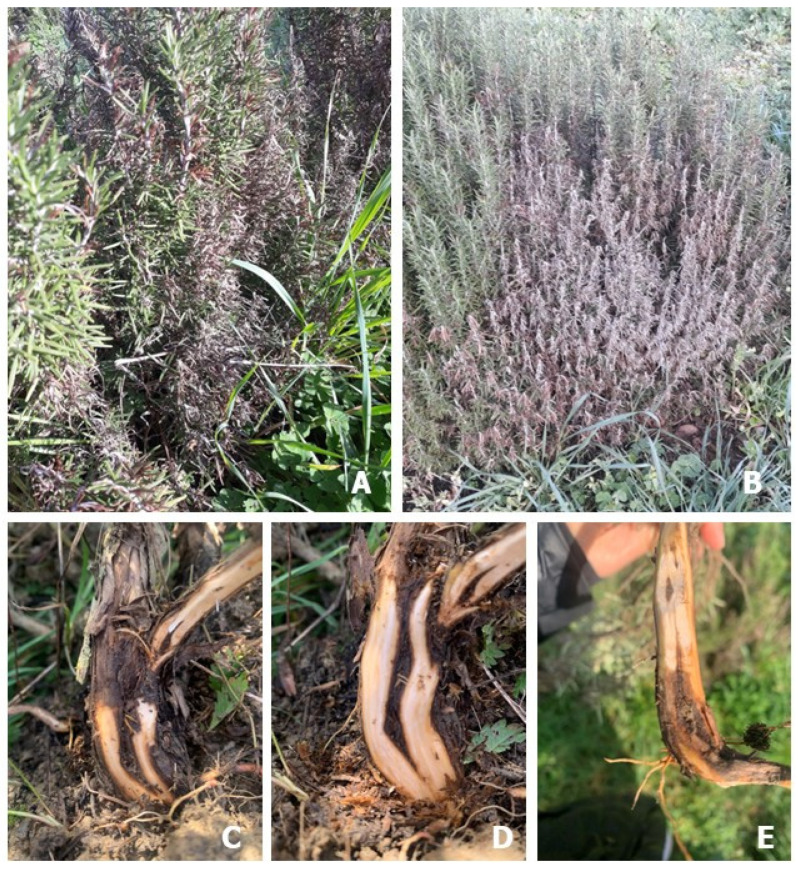
*Rosmarinus officinalis* individuals with decline/dieback symptoms in the field: (**A**) initial symptoms, with severe ash-grey discoloration/wilting; (**B**) dead (foreground) and dying plants in a row; (**C**) an affected basal stem portion, with internal tissue necrosis; (**D**) the same sample from the previous image (**C**) further debarked to ascertain the extent of underbark tissue necrosis; (**E**) uprooted and debarked plant for assessment of the extent of injury by *P. pseudocryptogea*.

**Figure 2 microorganisms-13-00567-f002:**
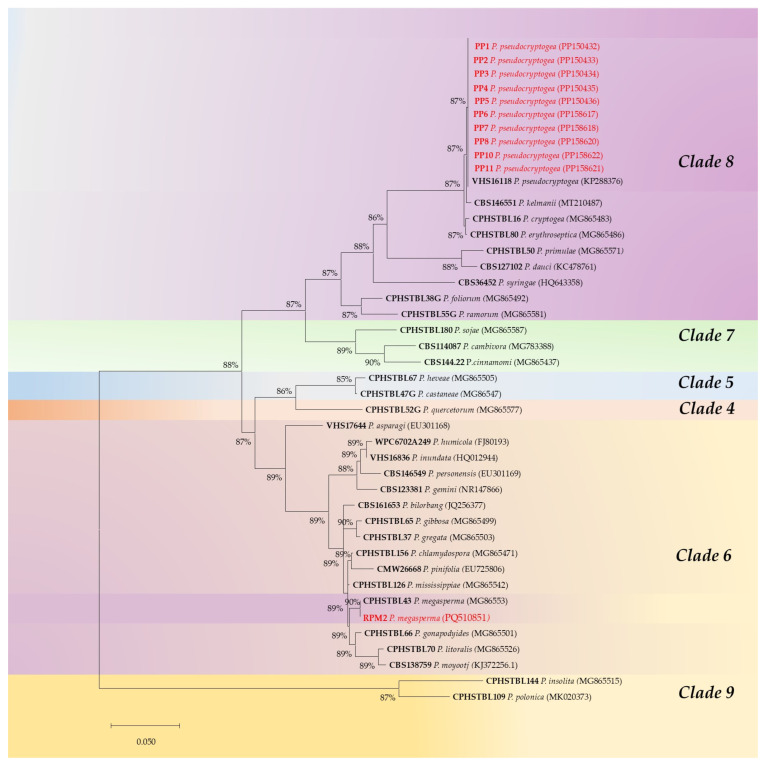
Maximum likelihood tree obtained from internal transcribed spacer (ITS) sequences of *Phytophthora* species representative of six clades. Data are based on the General Time Reversible model. A discrete Gamma distribution was used to model evolutionary rate differences among sites. The tree is drawn to scale, with branch lengths measured in the number of substitutions per site. Bootstrap support values in percentages (1000 replicates) are given at the nodes. Ex-type cultures are in bold; isolates obtained in this study are in red.

**Figure 3 microorganisms-13-00567-f003:**
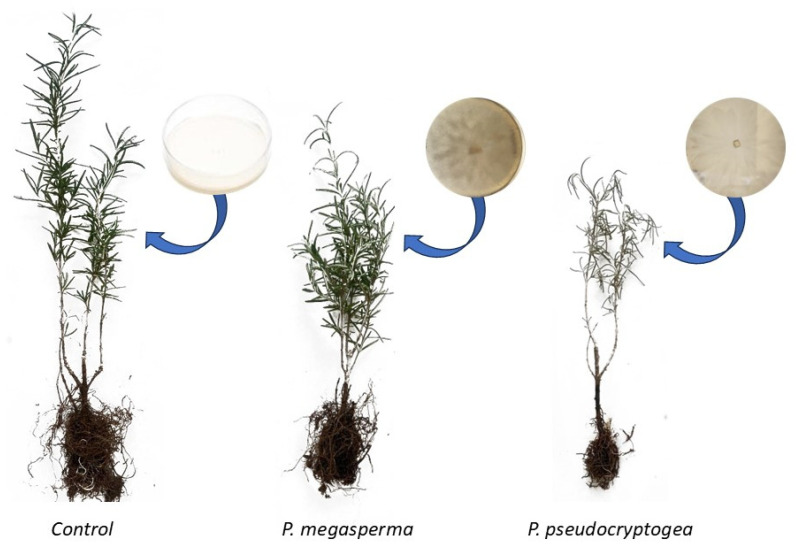
Visual representation of the outcome of pathogenicity tests on three *R. officinalis* plants 10 days post-inoculation. From left to right: a healthy plant inoculated with a sterile agar plug; a plant inoculated with *P. megasperma*; and a dead plant inoculated with *P. pseudocryptogea*.

**Figure 4 microorganisms-13-00567-f004:**
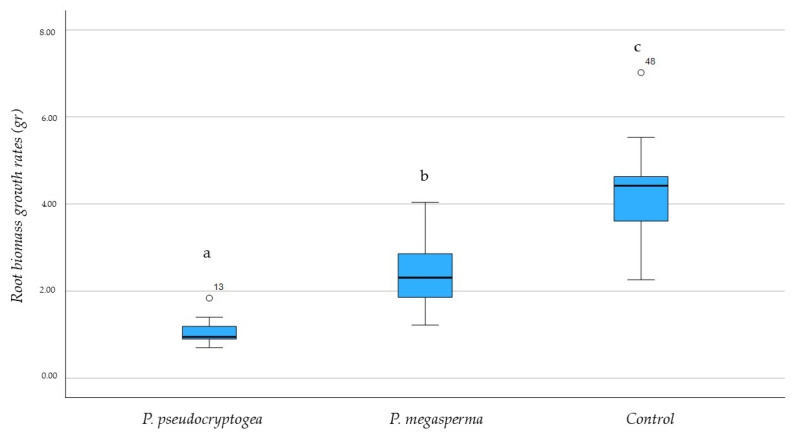
Box plots showing root biomass growth rates in *Rosmarinus officinalis* plants following artificial inoculation. Boxes represent the interquartile range (IQR), with medians indicated by horizontal lines, while whiskers extend to ±1.5 times the IQR. A one-way ANOVA test was performed to compare the mean root biomass growth (in grams) among treatments and the control, with statistical significance set at *p* < 0.05. Different letters indicate statistically significant differences between treatments.

**Table 1 microorganisms-13-00567-t001:** Isolates of *Phytophthora pseudocryptogea* and *Phytophthora megasperma* recovered from various matrices (root tissue, stem tissue, rhizosphere soil specimens) sampled at the *Rosmarinus officinalis* plantation in Casole d’Elsa, Tuscany, central Italy.

Isolate Code	Taxon	Source	Gene Bank Acc. No.
PP1	*P. pseudocryptogea*	Root tissue	*PP150432*
PP2	*P. pseudocryptogea*	Root tissue	*PP150433*
PP3	*P. pseudocryptogea*	Root tissue	*PP150434*
PP4	*P. pseudocryptogea*	Root tissue	*PP150435*
PP5	*P. pseudocryptogea*	Rhizosphere soil	*PP150436*
PP6	*P. pseudocryptogea*	Basal stem	*PP158617*
PP7	*P. pseudocryptogea*	Rhizosphere soil	*PP158618*
PP8	*P. pseudocryptogea*	Root tissue	*PP158620*
PP10	*P. pseudocryptogea*	Root tissue	*PP158622*
PP11	*P. pseudocryptogea*	Basal stem	*PP158621*
RPM2	*P. megasperma*	Rhizosphere soil	*PQ510851*

## Data Availability

The sequence data presented in this study were deposited in the NCBI GenBank data repository. *PP1*: *PP150432*; *P. pseudocryptogea*; https://www.ncbi.nlm.nih.gov/nuccore/PP150432; 22-JAN-2024; *PP2*: *PP150433*; *P. pseudocryptogea*; https://www.ncbi.nlm.nih.gov/nuccore/PP150433; 22-JAN-2024; *PP3*: *PP150434*; *P. pseudocryptogea*; https://www.ncbi.nlm.nih.gov/nuccore/PP150434; 22-JAN-2024; *PP4*: *PP150435*; *P. pseudocryptogea*; https://www.ncbi.nlm.nih.gov/nuccore/PP150435; 22-JAN-2024; *PP5*: *PP150436*; *P. pseudocryptogea*; https://www.ncbi.nlm.nih.gov/nuccore/PP150436; 22-JAN-2024; *PP6*: *PP158617*; *P. pseudocryptogea*; https://www.ncbi.nlm.nih.gov/nuccore/PP158617; 23-JAN-2024; *PP7*: *PP158618*; *P. pseudocryptogea*; https://www.ncbi.nlm.nih.gov/nuccore/PP158618; 23-JAN-2024; *PP8*: *PP158620*; *P. pseudocryptogea*; https://www.ncbi.nlm.nih.gov/nuccore/PP158620; 23-JAN-2024; *PP10*: *PP158622*; *P. pseudocryptogea*; https://www.ncbi.nlm.nih.gov/nuccore/PP158622; 23-JAN-2024; *PP11*: *PP158621*; *P. pseudocryptogea*; https://www.ncbi.nlm.nih.gov/nuccore/PP158621; 23-JAN-2024; *RPM2*: *PQ510851*; *P. megasperma*; https://www.ncbi.nlm.nih.gov/nuccore/PQ510851; 30-OCT-2024.

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
