# Peer review of "Morphological, Molecular and Pathological Characterization of Phytophthora pseudocryptogea Associated with Rosmarinus officinalis Dieback in Tuscany, Central Italy"

_microorganisms, 2025, doi:10.3390/microorganisms13030567_

Round 1
Reviewer 1 Report
Comments and Suggestions for Authors
The manuscript addressed Rosemary dieback in Italy and showed symptoms of root and crown rot. Phytophthora pseudocryptogea was identified as the causal agent, confirmed by isolation and inoculation studies. P. megasperma was also isolated from the soil but was non-pathogenic. The manuscript is generally well-structured; however, the following points require attention before acceptance.
1- The significance/conclusion of the study should be mentioned near the end of the abstract.
2-Condense the initial information about rosemary's uses and properties.
3-Start the introduction by addressing that rosemary can be affected by various pathogens. However, immediately emphasize Phytophthora as a particularly serious threat.
4- The novelty of the study should be highlighted near the end of the introduction.
5- Check the figure numbering (figure 2 is repeated) and fix the other relevent parts accordingly.
6-Elaborate more on the possible reasons/roles of P. megasperma presence rhizosphere soil at a low frequency (6%)? any ecological role in this system?
Author Response
Comments and Suggestions for Authors
The manuscript addressed Rosemary dieback in Italy and showed symptoms of root and crown rot. Phytophthora pseudocryptogea was identified as the causal agent, confirmed by isolation and inoculation studies. P. megasperma was also isolated from the soil but was non-pathogenic. The manuscript is generally well-structured; however, the following points require attention before acceptance.
- The significance/conclusion of the study should be mentioned near the end of the abstract.
Done.
- Condense the initial information about rosemary's uses and properties.
Done.
- Start the introduction by addressing that rosemary can be affected by various pathogens. However, immediately emphasize Phytophthoraas a particularly serious threat.
Done.
- The novelty of the study should be highlighted near the end of the introduction.
Done.
- Check the figure numbering (And with each oomycete infected separately) and fix the other relevent parts accordingly.
Done. Figure 2 is not repeated, it depicts the same specimen further debarked to ascertain the extent of necrosis. We clearly specified this in the caption
- Elaborate more on the possible reasons/roles of P. megasperma presence rhizosphere soil at a low frequency (6%)? any ecological role in this system?
Done.

Reviewer 2 Report
Comments and Suggestions for Authors
This manuscript contains valuable results and should be published in Microorganisms. This manuscript is devoted to rosemary (Rosmarinus officinalis) plants that showed symptoms of dieback in central Italy. Disease symptoms are described and isolations from diseased plants and from rhizosphere soil are made. Based on morphological and molecular analyses, two oomycetous species Phytophthora pseudocryptogea and P. megasperma were identified. Pathogenicity tests showed that P. pseudocryptogea is the cause of the dieback of rosemary plants. The research methods and results are presented in a clear and understandable way. However, it is advisable to provide some data in more detail in both of these chapters (see Remarks). The manuscript is written carefully, only a few minor errors were noticed, therefore the manuscript requires a minor revision.
Remarks
Line 13 stem or stems?
Line 93 should be clarified from how many plants were isolated and from how many soil samples
Line 193-194 Were symptoms observed on each of the following organs on each individual plant: shoots, crown, basal stem, collar roots, coarse roots and fine roots.
Line 208-209 ‘P. pseudocryptogea and P. megasperma, were consistently isolated from samples’ while in line 212 it is stated that only 1 isolate belonged to P. megasperma – this is a bit of a contradiction. This text requires some changes.
Line 215 ‘with a mean isolation frequency of 90%.’ It is unclear what this means? It would be good to provide specific data on how many isolations were made (number of plants, number of plant organs) and from how many samples Phytophthora was isolated
Line 226 ‘V8A, MEA and PDA media. ‘in Materials and Methods (section 2.2) it is not stated that colonies were grown on MEA and PDA media.
Line 233 ‘In dual cultures, all isolates produced gametangia’ – Materials and Methods it is not stated that dual cultures were performed. This should be completed.
Line 245 5.9μm – add a space
Line 267 infection or inoculation?
Line 283 infected or inoculated?
Line 273 change ‘Figure 2’ to ‘Figure 3’
Line 283 change ‘Figure 2’ to ‘Figure 3’
Line 275-276 ‘A reduction in growth rate and biomass production was observed in R. officinalis plants inoculated with both oomycetes;’ – it should be explained in Discussion, as P. megasperma did not cause symptoms
Line 305 ‘Phytophthora’ - P is not italic
Line 322 change ‘P. pseudcryptogea’ to ‘P. pseudocryptogea’
Line 432 Golovinomyces orontii - it should be italic (this also applies to other Latin names in References)
Author Response
Comments and Suggestions for Authors
This manuscript contains valuable results and should be published in Microorganisms. This manuscript is devoted to rosemary (Rosmarinus officinalis) plants that showed symptoms of dieback in central Italy. Disease symptoms are described and isolations from diseased plants and from rhizosphere soil are made. Based on morphological and molecular analyses, two oomycetous species Phytophthora pseudocryptogea and P. megasperma were identified. Pathogenicity tests showed that P. pseudocryptogea is the cause of the dieback of rosemary plants. The research methods and results are presented in a clear and understandable way. However, it is advisable to provide some data in more detail in both of these chapters (see Remarks). The manuscript is written carefully, only a few minor errors were noticed, therefore the manuscript requires a minor revision.
Remarks
Line 13 stem or stems?
Done.
Line 93 should be clarified from how many plants were isolated and from how many soil samples
Done, we made the sentence clearer.
Line 193-194 Were symptoms observed on each of the following organs on each individual plant: shoots, crown, basal stem, collar roots, coarse roots and fine roots.
Although with some variation in the severity of crown damage (some plants presented completely damaged crowns while others showed partial crown rot) symptoms were observed on all plant organs.
Line 208-209 ‘P. pseudocryptogea and P. megasperma, were consistently isolated from samples’ while in line 212 it is stated that only 1 isolate belonged to P. megasperma – this is a bit of a contradiction. This text requires some changes.
Done. Consistently has been deleted.
Line 215 ‘with a mean isolation frequency of 90%.’ It is unclear what this means? It would be good to provide specific data on how many isolations were made (number of plants, number of plant organs) and from how many samples Phytophthora was isolated
Done, the number of samples and isolates obtained has been specified above (line 250).
Line 226 ‘V8A, MEA and PDA media. ‘in Materials and Methods (section 2.2) it is not stated that colonies were grown on MEA and PDA media.
Done.
Line 233 ‘In dual cultures, all isolates produced gametangia’ – Materials and Methods it is not stated that dual cultures were performed. This should be completed.
Done. We described in the Materials and Methods that dual cultures were made ("pairing them with known A1 and A2 mating types") (line 154 and following lines)
Line 245 5.9μm – add a space
Done.
Line 267 infection or inoculation?
We prefer to keep the term inoculation because it more clearly indicates infection made by man: administration of infective material or a modified preparation of an infective agent (Oxford English Dictionary)
Line 283 infected or inoculated?
See answer in previous point
Line 273 change ‘Figure 2’ to ‘Figure 3’
Done.
Line 283 change ‘Figure 2’ to ‘Figure 3’
Done.
Line 275-276 ‘A reduction in growth rate and biomass production was observed in R. officinalis plants inoculated with both oomycetes;’ – it should be explained in Discussion, as P. megasperma did not cause symptoms
Done. We have added this information in Discussion
Line 305 ‘Phytophthora’ - P is not italic
Done.
Line 322 change ‘P. pseudcryptogea’ to ‘P. pseudocryptogea’
Done.
Line 432 Golovinomyces orontii - it should be italic (this also applies to other Latin names in References)
Done.

Reviewer 3 Report
Comments and Suggestions for Authors
Dear Authors,
Authors concentrated on Phytophthora pseudocryptogea effect on Rosmarinus officinalis.
Morphological and molecular identification (ITS and Cox1 sequencing) of strains isolated from symptomatic stem, roots and soil, revealed the occurrence of two Phytophthora species: Phytophthora pseudocryptogea, which constituted 94% of isolates obtained from the stem, roots and rhizosphere as well as P. megasperma which was not recovered from plant organs or tissue, but being exclusively isolated from rhizosphere soil samples.
In my humble opinion the strongest part of the manuscript is isolates identification and phylogenetic presentation. Unfortunately, the worse aspect was pathogenity test based almost only on symptoms reaction.
Introduction gives the reader sufficient background to analyzed obtained results;
Material and methods are quite clear presented in a repetitive way;
Discussion seems to be clear, but concentrated only on hosts plant and symptoms
Therefore, I presented a few important issues that it should be clarified an/or explained:
Please, rethink title reorganization/ change to be more adapted to achieved results.
- What does it mean root apparatus ?
- Please, define the aim of the studies in details;
- Pathogenicity of the obtained strains was assessed, but Authors stated that Plants inoculated with P. megasperma did not show any visible symptom -suggesting megasperma to be nonpathogenic – why Authors did not use molecular test specific for that types of reaction ? that statement in current form is too speculative and should be definitely confirmed.
- It is quite unusual that Authors used advanced molecular test like ITS, but molecularly not confirmed pathogenicity – this must be completed;
- Figure 3 – Authors should work on a data presentation;
Author Response
Comments and Suggestions for Authors
Dear Authors,
Authors concentrated on Phytophthora pseudocryptogea effect on Rosmarinus officinalis.
Morphological and molecular identification (ITS and Cox1 sequencing) of strains isolated from symptomatic stem, roots and soil, revealed the occurrence of two Phytophthora species: Phytophthora pseudocryptogea, which constituted 94% of isolates obtained from the stem, roots and rhizosphere as well as P. megasperma which was not recovered from plant organs or tissue, but being exclusively isolated from rhizosphere soil samples.
In my humble opinion the strongest part of the manuscript is isolates identification and phylogenetic presentation. Unfortunately, the worse aspect was pathogenity test based almost only on symptoms reaction.
Introduction gives the reader sufficient background to analyzed obtained results;
Material and methods are quite clear presented in a repetitive way;
Discussion seems to be clear, but concentrated only on hosts plant and symptoms
Therefore, I presented a few important issues that it should be clarified an/or explained:
Please, rethink title reorganization/ change to be more adapted to achieved results.
Done, the title has been reformulated
- What does it mean root apparatus ?
The term "apparatus" has been replaced with "system" (e.g. "root system").
- Please, define the aim of the studies in details;
Done
- Pathogenicity of the obtained strains was assessed, but Authors stated that Plants inoculated with P. megasperma did not show any visible symptom -suggesting megasperma to be nonpathogenic – why Authors did not use molecular test specific for that types of reaction? That statement in current form is too speculative and should be definitely confirmed.
We have clarified the results of our inoculation tests in our study trying to better define the ecological role of Phytophthora megasperma
- It is quite unusual that Authors used advanced molecular test like ITS, but molecularly not confirmed pathogenicity – this must be completed;
Thanks for this suggestion. In fact we had not highlighted that (as already reported for P. pseudocryptogea) also P. megasperma has been identified with morphological and molecular test (ITS)
- Figure 3 – Authors should work on a data presentation;
Done. The caption for Figure 3 (which in this R1 version has become Figure 4) has been reworded and made clearer.
